# Analysis of Circulating C19MC MicroRNA as an Early Marker of Hypertension and Preeclampsia in Pregnant Patients: A Systematic Review

**DOI:** 10.3390/jcm11237051

**Published:** 2022-11-29

**Authors:** Adrianna Kondracka, Ilona Jaszczuk, Dorota Koczkodaj, Bartosz Kondracki, Karolina Frąszczak, Anna Oniszczuk, Magda Rybak-Krzyszkowska, Jakub Staniczek, Agata Filip, Anna Kwaśniewska

**Affiliations:** 1Department of Obstetrics and Pathology of Pregnancy, Medical University of Lublin, 20-059 Lublin, Poland; 2Department of Cancer Genetics with Cytogenetic Laboratory, Medical University of Lublin, 20-059 Lublin, Poland; 3Department of Cardiology, Medical University of Lublin, 20-059 Lublin, Poland; 4Department of Oncological Gynecology and Gynecology, Medical University of Lublin, 20-059 Lublin, Poland; 5Department of Inorganic Chemistry, Medical University of Lublin, 20-059 Lublin, Poland; 6Department of Obstetrics and Perinatology, University Hospital, 31-501 Krakow, Poland; 7Department of Gynecology, Obstetrics and Gynecologic Oncology, Medical University of Silesia, 40-055 Katowice, Poland

**Keywords:** C19MC microRNA, early pregnancy biomarkers, hypertension, preeclampsia

## Abstract

Preeclampsia and hypertension complicate several pregnancies. Identifying women at risk of developing these conditions is essential to establish potential treatment modalities. Biomarkers such as C19MC microRNA in pregnant patients wopuld assist in defining pregnancy surveillance and implementing interventions. This study sought to analyze circulating C19MC microRNA as an early marker of hypertension and preeclampsia in pregnant patients. A systematic review was undertaken using the following registers: disease registries, pregnancy registries, and pregnancy exposure registries, and the following databases: PubMed, CINAHL, Web of Science, Scopus, and EMBASE. The risk of bias was assessed using the Cochrane technique. From the 45 publications retrieved from the registers and databases, only 21 were included in the review after the removal of duplicates, screening, and eligibility evaluation. All 210 publications had a low risk of bias and illuminated the potential use of circulating C19MC microRNA as an early marker of hypertension and preeclampsia in pregnant patients. Therefore, it was concluded that C19MC microRNA can be used as an early marker of gestational preeclampsia and hypertension.

## 1. Introduction

MicroRNAs have been suggested as possible hypertension and pre-eclampsia indicators since they are crucial cell process regulators. Most investigations have conducted the primate-specific microRNA cluster on chromosome 19 (C19MC microRNA) profiling analysis on total serum samples or maternal plasma to treat the later incidence of pregnancy-related problems, such as gestational pre-eclampsia, hypertension, and fetal growth restriction. Exosomal nanoparticles released into the blood and extracellular space include microRNAs [1]. They allow communication between close-by and far-off cells. Over the past decade, interest in forming non-invasive modes of cell-free nucleic acid detection has been on an upward trajectory. They include microRNAs during maternal circulation [2]. The ability to diagnose via given molecular biomarkers alongside amalgamating them into current prognosis algorithms for issues linked to pregnancy is vital [2]. Small non-coding RNAs (sncRNAs) guide post-transcriptional gene expression by blocking messenger RNA targets from translation. This systematic literature review analyzes findings from different sources on circulating C19MC microRNA as an early marker of hypertension and pre-eclampsia in pregnant patients. It establishes that circulating C19MC microRNAs may contribute to developing pre-eclampsia and prenatal hypertension in early pregnancy.

## 2. Materials and Methods

The review followed the PRISMA rule to report the stepwise procedure used to retrieve information from various databases and registers. The PRISMA guidelines were also adhered to strictly to eliminate bias and ensure the successful completion of the systematic literature review. Figure 1 below shows the PRISMA chart demonstrating various phases of the review.

The inclusion criterion required using systematic reviews investigating circulating C19MC microRNA as an early marker of hypertension and pre-eclampsia in pregnant patients. Other inclusion requirements included any study, be it experimental, cohort, or case study, articles published in English between 1 January 2013, and 5 August 2022, and original research undertaken in any region of the world with a sample size of at least five participants. The exclusion criteria were papers with fewer than 5 cases, papers published before 2013, and review papers except for systematic reviews. The exclusion criteria required the removal of studies from the review, encompassing articles published in languages other than English and materials that did not concentrate on investigating circulating C19MC microRNA as an early marker of hypertension and pre-eclampsia in pregnant patients.

### 2.1. Information Sources

The databases used in the review included PubMed, CINAHL, Web of Science, Scopus, and EMBASE. These databases were consulted simultaneously within one month (May 2022) to identify studies that could be included in the review. Similarly, registers such disease registries, pregnancy registries, and pregnancy exposure registries were also searched to identify publications aligned with the topic of interest. These registers were also searched simultaneously within one month (June 2022). The reference lists of the articles obtained from the databases and registers were also used to identify studies that focused on investigating various aspects of the topic of interest, circulating C19MC microRNA as an early marker of hypertension and pre-eclampsia in pregnant patients. The studies were also retrieved from the abovementioned databases and stored for subsequent processes and steps.

### 2.2. Search Strategy

The scholarly materials retrieved from the databases and registers mentioned above were limited to those published between 1 January 2013, and 5 August 2022. The keywords included C19MC microRNA, early hypertension markers, early pre-eclampsia markers, hypertension in pregnant patients, pre-eclampsia in pregnant patients, and C19MC microRNA in pregnant patients. A manual search was conducted by reading the bibliographies of the review articles or materials that were retrieved from the reference lists, and frequently mentioned publications on gestational hypertension and pre-eclampsia were used to uncover papers that were not identified by the electronic search.

### 2.3. Selection Process

The selection process was undertaken using three crucial steps. First, reviewers who worked independently selected all articles that were retrieved from the databases and registers to reduce the chances of bias. Second, all the randomly selected publications were reviewed to determine their eligibility. Their abstracts and titles were screened against the eligibility criteria to determine whether the articles that met the inclusion criterion. RobotAnalyst was used as an automation tool during the eligibility screening. This second step enabled the removal of duplicate publications and ushered in the last phase of the selection criterion, which mainly handled the articles with titles and abstracts that did not give sufficient information regarding the study. The last step involved a full-content evaluation by the reviewer to assert whether those particular publications could be included in the review. This three-step process was undertaken independently to ensure that only the necessary publications were included in the review.

### 2.4. Data Collection Process

The search yielded thirty articles about circulating C19MC microRNA, gestational hypertension, or pre-eclampsia. The threshold index was assessed before synthesizing the data. The diagnostic index tests, including summary receiver operating characteristic (SROC), diagnostic odds ratio (DOR), negative or positive likelihood ratio (NLR or PLR), specificity (Spe), and sensitivity (Sen), were measured with a confidence interval of 95%.

### 2.5. Data Items

The data items were manually extracted from the publications included in the review. The essential information from the selected reports was amassed and recorded in a table. Since this review focused on analyzing the circulating C19MC microRNA as an early marker of hypertension and pre-eclampsia in pregnant patients, the data collected from the selected articles included the author(s), titles, years of publication, and outcomes of the reports. The results segment of the chosen publications provided information concerning circulating C19MC microRNA as an early marker of gestational pre-eclampsia and hypertension. For all statistical studies, the statistical analysis tool or technique used and the statistical results were recorded under the data item and outcomes of the reports.

### 2.6. Study Risk of Bias Assessment

The Cochrane technique was used to assess the risk of bias in all the publications included in the review. Cochrane is a standard risk appraisal instrument that uses judgments of unclear risks (?), high risk (-), and low risk (+) on different axes for studies such as systematic reviews, which may have biases in their decisions, results, strategies, and other aspects of interest. One reviewer independently assessed the risk of bias in each study using the Cochrane tool.

### 2.7. Synthesis Method

The data in this review were synthesized using thematic analysis and grouping similar information, all presented in Table 1. The four columns included the publications’ author(s), titles, publication years, and outcomes. Under the outcomes column, the studies’ results were thematically presented, focusing on whether C19MC microRNA can be used as an early marker of gestational hypertension and pre-eclampsia. The rows comprised the heading row and the thirty studies included in the review.

### 2.8. Reporting Bias Assessment

The bias risk assessment performed using the Cochrane technique was reported using the Cochrane bias risk assessment, as shown in Table 2, which has eight columns and thirty-one rows. The columns included the study of interest, selection bias (random sequence generation), selection bias (allocation concealment), performance bias, detention bias, attrition bias, reporting bias, and other biases. The first row provides the heading information, while the other thirty comprise the publications included in the review.

## 3. Results

### 3.1. Study Selection

The results of this review can be categorized according to PRISMA guidelines. The search, identification, and retrieval of articles resulted in a collection of fifty-five publications; ten were retrieved from the databases, and forty-five were from the registers. The fifty-five publications were then checked, and five duplicates were removed. Two articles were removed because of their ineligibility, as marked by RobotAnalyst. Three publications were also removed because they were written in languages other than English. Only thirty-five remaining reports were subjected to screening. The screening process involved checking and evaluating their abstracts and titles to determine their suitability to be included in the review. One publication was excluded because its abstract did not include all the crucial keywords required. The remaining thirty-four articles were sought for retrieval, but only thirty-two were obtained because two publications were inaccessible. The thirty-two reports were assessed for eligibility. Two did not meet the eligibility criteria because one had less than five participants, while the other focused on C19MC microRNA as an early marker of pregnancy-related problems other than pre-eclampsia and hypertension. The remaining thirty publications met the eligibility criteria and were included in the review.

### 3.2. Study Characteristics

The study characteristics obtained in this review were divided into four groups: author(s), title, year of publication, and outcomes of the reports, as evident in Table 1. Table 1 shows that only one article was published by one author [2]. All other twenty-nine studies were published by two or more authors. All the publications had different titles directly associated with the research topic. Furthermore, they were published during different periods, as evident via their years of publication. Regarding year of publication, 3.33% of the articles were published in 2013 [1], 2014 [19], or 2015 [13], and 6.67% of the reports were published in 2016 [22,23] and 2019 [2,12]. In addition, 10% of the studies were published in 2017 [4,24,25] and 2018 [15,21,26], while 13.33% of the articles were published in 2021 [3,8,14,17] and 2022 [6,10,27,28]. Finally, 30% of the studies were published in 2020 [5,7,9,11,16,18,20,29,30]. This statistic shows the diversity of the documents’ retrieval where their publication years are concerned.

### 3.3. Risk of Bias in Studies

The results of the Cochrane bias risk assessment are evident in Table 2, which illuminates that all the materials have low reporting and selection bias risks and a high-performance bias risk. A significant number of the other Cochrane method measures underscored low bias risk. These results suggest that all the publications were of good quality and deserved to be included in the review.

### 3.4. Results of Individual Studies

#### 3.4.1. miRNAs of Different Stages of Preeclampsia

This review discovered informative facts about the role of C19MC microRNA as an early marker of gestational hypertension and pre-eclampsia. For instance, clinically confirmed hypertension and pre-eclampsia have been linked to changes in extracellular microRNA expression [13]. No distinction between fetal and normal pregnancies could be made on the basis of the levels of circulating microRNA expression. Furthermore, according to He and Ding, pre-eclampsia often appears after 20 weeks of pregnancy and is characterized by proteinuria and gestational or chronic hypertension [6]. The condition arises from a problem with placentation, which leads to insufficient uteroplacental blood perfusion and ischemia. Pre-eclampsia is an implantation condition, and its reasons are yet unclear. Its fundamental etiological theory assumes that placentation and insufficient trophoblast invasion are related to a poor adaptation of the local maternal immune system to extra-villous cytotrophoblast produced at the fetal–maternal interface [19].

Czernek and Duchler examined the C19MC microRNA gene expression in simple and complex pregnancies. According to their assertion, there are 56 microRNA genes in the chromosome 19 microRNA cluster. The research focused primarily on microRNAs previously indicated to be substantially present in placental tissues and those reported to be uniquely expressed in them. The research examined C19MC microRNA gene expression in connection with established risk factors for worse perinatal results. Analysis of C19MC microRNA gene expression connected to the degree of clinical symptoms, the doppler ultrasonography parameters, and the delivery date was conducted to determine the severe condition. Complex pregnancies and controls generally have distinct expression profiles for C19MC microRNAs. Pre-eclampsia patients had down-regulation of C19MC microRNAs more often in specific subgroups of pregnancy-related disorders. Further results demonstrated that one C19MC microRNA (miR-517-5p) was changed in pre-eclampsia, necessitating abortion before the nine month gestation period, while five C19MC microRNAs (miR-26b-5p, miR-7-5p, miR-181a-5p, hsa-miR-486-1-5p, and hsa-miR-486-2-5p) in severe preeclamptic pregnancies were dysregulated. The findings imply that the microRNAs have a role in pre-eclampsia’s pathophysiology [29]. 

Additionally, moderate pre-eclampsia that lasted for a few weeks and was closely followed from the time of diagnosis to birth was shown to have a similar expression pattern of placental-specific microRNAs [22]. However, the down-regulation of placental-specific microRNAs seemed to be more profound the longer the pregnancy-related illness persisted. This implies that certain C19MC microRNA dysregulation may represent a compensatory strategy rather than the disease process itself. In another study, Whigham et al. discovered maternal plasma C19MC microRNAs that distinguish between healthy pregnancies and non-pregnancies [20]. More recent research has shown that established pre-eclampsia is characterized by an increase in the circulating miR-526a, miR-525, miR-520a-5p, miR-517-5p, and miR-516-5p genes. While the group of patients with hypertension alongside the control did not have different plasma levels of microRNAs, elevated levels were seen in the group of participants with developed pre-eclampsia [20].

Whigham et al. also note the capacity of extracellular C19MC microRNAs to distinguish between complicated and normal pregnancies at the beginning of pre-eclampsia with or without fetal growth restriction, which was verified using both relative and absolute quantification methods. Unfortunately, there is a lack of information comparing extracellular C19MC microRNA levels in abnormal and normal pregnancies [20]. The study’s findings disagree with MacDonald et al., who noted an increase of extracellular miR-520h in four pre-eclampsia patients. The findings prompted an additional investigation into the relationships between circulating C19MC microRNAs and illness severity concerning the degree of delivery needs and clinical symptoms [28].

Furthermore, no correlation among the gene, plasmatic expression, and risk factors for lower C19MC microRNA neonatal levels was found in the association investigation. In both pregnancies with moderate and severe pre-eclampsia, the microRNA plasmatic and gene expression levels were comparable. According to the study’s findings, many pathological and physiological processes depend heavily on microRNAs and are to blame for pregnancy-related problems. Consequently, circulating C19MC microRNAs may have a role in the etiology of pre-eclampsia [7]. The research involved increased circulating C19MC microRNAs that characterize the intriguing discovery that developed pre-eclampsia.

##### miRNAs in the Normal and Abnormal Pregnancies

Even though C19MC microRNAs appear to be down-regulated in the placental tissues in response to a variety of pregnancy-related disorders, including pre-eclampsia and hypertension, as noted above, upregulation of the specific microRNAs appears only in the maternal circulation in pre-eclampsia cases [3]. The contradictory result may be interpreted in several different ways. This recent research by Ali et al. showed that, compared with gestation-matched controls, numerous microRNAs regulated by hypoxia were upregulated in pregnancies affected by significant preterm fetal growth limitation [3]. However, most studies focused on examining pregnancy-related microRNAs whose genes are not found in the miRNA clusters on the chromosome.

A large number of the physiological changes that occur during a typical pregnancy are caused by an acute-phase reaction that is triggered by an inflammatory response. The proximal source of the issues is the placenta [4]. Apoptotic bodies or syncytiotrophoblast microparticles, which include fetus and placenta extracellular nucleic acids, are released by the placenta while it undergoes continuous remodeling throughout normal pregnancy. Cronqvist et al. contended that circulating syncytiotrophoblast exosomes contribute to some syndrome symptoms and maternal inflammation. Pre-eclampsia (PE), hypertension, and other diseases’ clinical characteristics may be explained by increased inflammation-related symptoms that are reportedly present in healthy pregnancies at term [4].

According to Jin et al., pre-eclampsia with a clinically confirmed diagnosis is linked to changes in extracellular microRNA expression. However, after the evaluation of circulating microRNA, there was no difference between normal and abnormal pregnancies [8]. Špačková found that several hypoxia-regulated microRNAs were complicated by extremely preterm fetal growth restriction [2]. Nonetheless, most researchers concentrated on investigating microRNAs connected with pregnancy whose genes are not included in the chromosome 19 miRNA clusters or C14MC.

Additionally, Qin et al. further discovered that C19MC microRNAs are present in maternal plasma and have been shown to distinguish between healthy pregnancies and non-pregnancies. They observed increased extracellular C19MC microRNA levels in regularly developing pregnancies. The findings of a follow-up study demonstrated that the up-regulation of the C19MC microRNAs is a defining feature of pre-eclampsia that has already developed [17]. Srinivasan et al. indicated the ability of extracellular C19MC microRNAs to distinguish between individuals at risk of later developing placental-sufficiency-related issues and normal pregnancies at the onset of gestation. The findings emphasized the necessity for further investigation of extracellular microRNAs in maternal circulation with the goal of regular evaluation in daily practice and identification as possible indicators for pregnancy complications linked to placental insufficiency [18].

In addition, Jing et al. asserted that even though fetal growth restriction and pre-eclampsia may be detected using separate serum indicators or maternal plasma, combination screening tests are presently employed in clinical settings to determine the likelihood of developing pre-eclampsia. Pregnancy-associated plasma protein-A, placental growth factor, and, together with maternal blood biomarkers, uterine artery Doppler and maternal risk factors may detect roughly 95% of patients with early-onset pre-eclampsia with a 10% false-positive rate. [9] To enhance the prediction of problems associated with placental insufficiency, additional studies are required to find additional biomarkers with higher diagnostic performance. Jing et al. examined 754 miRNAs and found no predictive value for early pre-eclampsia in first-trimester maternal blood miRNA evaluation. According to miRNA profiling using the high-throughput Open Array TM technology, seven microRNAs have a distinct abundance profile in early pre-eclampsia. Hence, there were no discernible changes between pre-eclampsia and controls after validation by real-time quantitative analysis [9].

#### 3.4.2. miRNA as a Biomarker of Gestational Hypertension

To determine whether a combination of miR-518b and miR-520h biomarkers or a single plasmatic miR-520h biomarker offers valuable tools in the risk assessment for gestational hypertension, several large-scale, multi-center studies, including individuals from various demographics, are required [25]. The rise in extracellular C19MC microRNAs during the first trimester of pregnancy may be related to the down-regulation of various hormones and proteins studied as potential early markers for pre-eclampsia and hypertension. Various diseases, such as gestational hypertension, are associated with placental exosomes’ content during pregnancy [27]. Wommack et al. focused on examining the role of miRNAs as signaling molecules. The authors discovered that miRNAs operate as co-regulated groups of signaling molecules to coordinate infant outcomes and gestation length [21].

However, using quantitative RT-PCR to test 30 non-placental microRNAs in maternal mononuclear cells from peripheral blood, Mavreli et al. accurately predicted late pre-eclampsia and miscarriage during the first trimester of pregnancy. The results were assessed with the help of a designed system, awarding points to each participant whose microRNA quantification outcome fell within the top eight. Findings were arranged from the highest to the lowest Ct value for each microRNA. Each patient’s unique pregnancy risk score was calculated once the findings of all microRNAs were added together. Four microRNAs had very low values; therefore, they were deemed technically unfit for analysis and were not included in the score [26]. Similarly, Špačková used microarray analysis to find 19 mature miRNAs that were differentially expressed in the blood of pregnant women who later had acute pre-eclampsia. Of them, 12 were upregulated, and 7 were down-regulated during the early gestational phases. In the blood of women who eventually experienced acute pre-eclampsia, mir-1233 was the most overexpressed [2].

In addition, to identify C19MC microRNAs with extracellular placental specificity in maternal circulation as possible biomarkers for problems associated with placental insufficiency, Li et al. suggested the necessity for a more thorough investigation of the microRNAs. Using the absolute and relative quantification methods, the ability of extracellular C19MC microRNAs to distinguish between normal pregnancies and people predisposed to develop intrauterine growth restrictions and pre-eclampsia in early pregnancy was described [11]. The pilot study involved six pregnant women with one early IUGR, four late pre-eclampsias, and one early pre-eclampsia [5]. As the findings revealed that the females who subsequently had severe pre-eclampsia had upregulated placental-specific miR-520a in their sera at 12–14 weeks of pregnancy, Li et al. considerably contributed to validating the findings by Munjas et al. [14]. From the stem-loop and miR-520a* combined, mir-520a (miR-520a-3p) is produced. Akin as well as Li et al. and Hornakova et al. found that circulating miR-517* was upregulated in preeclampsia-prone early pregnancy [11,14,30].

Consequently, subsequent large-scale investigations are required to evaluate the positive predictive value, specificity, and sensitivity of C19MC microRNAs for hypertension or pre-eclampsia. The effectiveness of placental-specific microRNAs for diagnosing disease severity should be assessed in connection with Doppler ultrasonography characteristics, delivery needs, and clinical symptoms [10]. The research produced intriguing results, showing that up-regulation of circulating C19MC microRNAs is a hallmark of early pregnancy, which is predisposed to developing issues linked to gestational hypertension and placental insufficiency. In addition, elevated plasmatic levels of miR-516-5p, miR-518b, and miR-520h in the first trimester alone are strong indicators of future gestational hypertension. One C19MC placental-specific microRNA biomarker may be used to screen for the start of hypertension in the first trimester of pregnancy [23]. Alternately, miR-518b and miR-520h, both placental-specific C19MC microRNA prediction biomarkers, may be combined to forecast the incidence of prenatal hypertension.

#### 3.4.3. Possibilities of Using miRNA in Clinical Diagnostics

Although combinations of second-trimester biochemical indicators, markers, and ultrasonography have been proposed, none has yet shown findings that are sound enough to be used in a therapeutic setting [19]. In the first trimester of maternal serum/plasma samples collected from patients with hypertension or pre-eclampsia, a number of the hypothesized targets of C19MC microRNAs in which the current study was interested were previously shown to be enhanced [10]. Several miRNAs may control the same gene. It is feasible to fully pinpoint the ones responsible for regulating specific genes of interest. Unfortunately, it is difficult to directly interpret experimental outcomes since complex networks typically govern the routes. Most of them target many genes for repression and collectively control them [16]. Hence, as mentioned earlier, pregnancy-related difficulties are caused by several pathological and physiological processes in which microRNAs play a vital role. The literature shows that circulating C19MC microRNAs may contribute to developing pre-eclampsia and prenatal hypertension in early pregnancy.

Furthermore, a different study by Légaré et al. did not find any information about “C19MC microRNA profiling in maternal plasma exosomes during the first trimester of pregnancy” [10]. Instead, it discovered that placental tissues generated from individuals with gestational hypertension and pre-eclampsia after childbirth had the same C19MC microRNA expression profile as first-trimester circulating plasma exosomes. Four of the fifteen C19MC microRNAs that were examined showed down-regulation in placental tissues when GH patients were present [15]. It is similar to the results in placental tissues taken from individuals with gestational hypertension at birth in patients with subsequent occurrences of hypertension. It was discovered after examining the first-trimester maternal plasma exosome C19MC microRNA expression profile of pregnancy, with the selection of only those with diagnostic potential.

In addition, eleven of the fifteen C19MC microRNAs evaluated showed down-regulation in pre-eclampsia patients. Légaré et al. discovered lower levels of miR-525-5p, miR-520a-5p, and miR-517-5p in individuals who subsequently developed pre-eclampsia, broadly matching the expression patterns reported in afflicted placental tissues [10]. The finding are in line with what was revealed in the researcher’s previous study. The microRNAs were tested for their diagnostic potential during the first trimester in maternal plasma exosomes of pregnancy. However, the findings are at odds with those of He and Ding, which showed that circulating C19MC microRNAs in maternal plasma were upregulated in the first trimester and accurately predicted the eventual onset of pre-eclampsia or gestational hypertension [6]. From 12 to 14 weeks of pregnancy, other researchers noticed elevated levels of several C19MC microRNAs in sera of women who eventually had severe pre-eclampsia [10]. Hence, a combination of variables, including those resulting from several different circumstances, may affect the different expression patterns of C19MC microRNAs between maternal plasma and their exosomes. At the very least, a representation of a specific C19MC microRNA expression in maternal plasma can be seen in placental cells from different regions undergoing apoptosis [24]. It releases placental debris into the mother’s bloodstream and actively secretes exosomes that promote intercellular communication.

## 4. Discussion

Research by Hromadnikova et al. revealed no correlation between microRNA and a history of hypertension among individuals with pre-eclampsia that had already developed. The expression of microRNA genes in placental tissues did not alter Doppler ultrasonography parameters linked to worse outcomes in pre-eclampsia. The elevation of pertinent proteins involved in the direction of critical biological processes, including hemocoagulation, apoptosis, stress response, and angiogenesis, may result from the lowered amounts of C19MC microRNAs in placental tissues [1]. Lv et al. further argued that ischemia, hypoxia, insufficient uteroplacental blood perfusion, and defective placental angiogenesis may all lead to blood coagulation–fibrinolysis system failure, aberrant placental trophoblast apoptosis, and, lastly, the emergence of a widespread maternal inflammatory response. Predicted targets of C19MC microRNAs are elevated in placental tissue samples from women who had problems during pregnancy [12].

However, the observed down-regulation of C19MC microRNAs is at odds with the lower levels of several proteins found in the tissue of the placenta in patients with problems in their pregnancies. According to Miura et al., the microRNAs are anticipated to be targeted. There exist modes of fully identifying miRNAs that guide specific genes of value [13]. However, their routes are often complicated control networks that are hard to grasp. Moreover, it complicates the straightforward interpretation of experimental results [27]. Many target several genes for suppression, and they seem collaboratively regulated.

Lastly, the previous theory that exosomes discharged into the body’s circulation serve as a non-invasive and singular source of signaling molecules, whose abnormal expression profile mimics that of the parent cells, was supported by this review. It lends credence to the hypothesis that those produced by the placenta may be used in first-trimester screening to detect a sizable fraction of women who may later develop pre-eclampsia or gestational hypertension [21]. The only drawback of the strategy is that since the down-regulation of the same biomarkers begins early in pregnancy, the screening of C19MC microRNAs in plasma exosomes cannot distinguish between the women who will later develop hypertension and those who will have pre-eclampsia during the first trimester of pregnancy. Nevertheless, it may one day lead to the discovery of new microRNA biomarkers that may distinguish between women at risk for gestational hypertension or pre-eclampsia, allowing for the early determent of pre-eclampsia with earlier delivery of low-dose aspirin.

## 5. Conclusions

In conclusion, this literature review showed that circulating C19MC microRNAs may contribute to developing pre-eclampsia and prenatal hypertension in early pregnancy. In individuals with subsequent occurrences of gestational hypertension and pre-eclampsia, C19MC microRNAs were shown to be down-regulated. The circulating C19MC microRNA expression profile from the first trimester was identical to that in placental tissues collected from individuals with hypertension and pre-eclampsia. Expression analysis of maternal plasma exosomes, as opposed to entire maternal plasma samples, increased the first trimester C19MC microRNA screening’s prediction accuracy for detecting hypertension and pre-eclampsia. The results require further confirmation by large-scale investigations. More first-trimester plasma samples must be gathered to achieve a sufficient number of individuals who may later suffer pregnancy-related problems, making conducting the study very difficult.

## Figures and Tables

**Figure 1 jcm-11-07051-f001:**
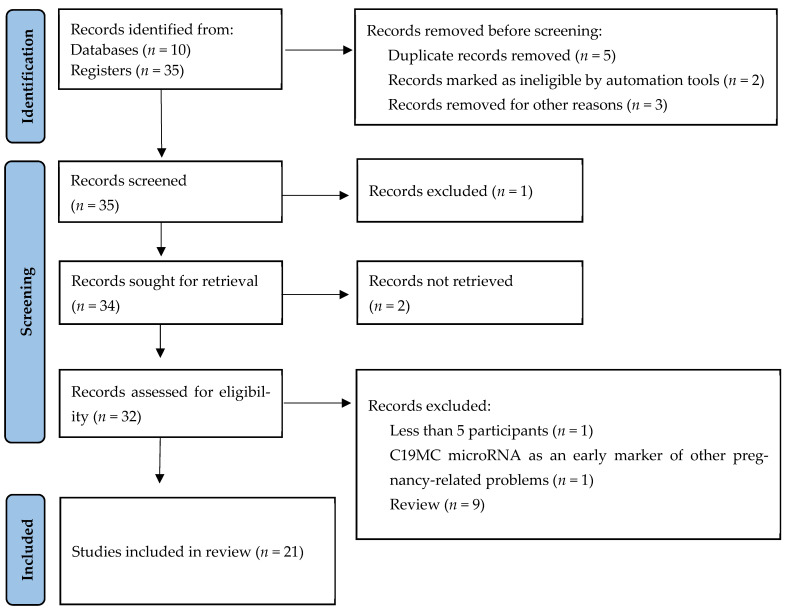
PRISMA chart showing the different stages of the systematic literature review. Ten records were retrieved from databases: PubMed, CINAHL, Web of Science, Scopus, and EMBASE. Thirty-five records were retrieved from registers: disease registries, pregnancy registries, and pregnancy exposure registries. Before the screening, 5 duplicate records were removed, 2 were marked as ineligible using RobotAnalyst, and 3 records were removed because they were written in languages other than English. Thirty-five records were screened, and one was excluded, as its abstract did not include all the crucial keywords required. Thirty-four records were sought for retrieval and only thirty-two were retrieved. Two records could not be retrieved. The 32 records were assessed for eligibility and 2 records were excluded because 1 had less than five participants, while the other focused on C19MC microRNA as an early marker of pregnancy-related problems other than pre-eclampsia and hypertension and nine were review. Therefore, 21 records were included in the review.

**Table 1 jcm-11-07051-t001:** Study Characteristics.

Author(s)	Title	Year	Sample Size	Source	Method	Outcomes
Ali, Asghar et al. [3]	MicroRNA–mRNA Networks in Pregnancy Complications: A Comprehensive Downstream Analysis of Potential Biomarkers	2021	127	Maternal circulation/Placenta	Real-time PCR	Upregulation of microRNAs appears only in the maternal circulation in pre-eclampsia cases
Cronqvist Tina et al. [4]	Syncytiotrophoblast Derived Extracellular Vesicles Transfer Functional Placental Mirnas to Primary Human Endothelial Cells	2017	10	Placental cotyledons	Real-time PCR	Circulating syncytiotrophoblast debris contributes to some of the symptoms of maternal inflammation. Pre-eclampsia (PE), hypertension, and other diseases’ clinical characteristics may be explained by an increase in inflammation-related symptoms that are reportedly present in healthy pregnancies at term
Demirer, Selin et al [5]	Expression Profiles of Candidate MicroRNAs in the Peripheral Blood Leukocytes of Patients with Early- and Late-Onset Preeclampsia versus Normal Pregnancies	2020	148	Maternal blood samples	Real-time PCR	The pilot study involved six pregnant women who had one early IUGR, four late pre-eclampsias, and one early pre-eclampsia
He, Xin, and Dan-Ni Ding [6]	Expression and Clinical Significance of Mir-204 in Patients with Hypertensive Disorder Complicating Pregnancy	2022	196	Maternal peripheral blood	Real-time PCR	Preeclampsia often appears after 20 weeks of pregnancy and is characterized by proteinuria and gestational or chronic hypertension
Hromadnikova, Ilona et al. [1]	Circulating C19MC microRNAs in Preeclampsia, Gestational Hypertension, and Fetal Growth Restriction	2013	113	Maternal peripheral blood	Real-time PCR	No correlation exists between microRNA and a history of hypertension among individuals with pre-eclampsia that had already developed
Jelena, Munjas et al. [7]	Placenta-Specific Plasma miR518b is a Potential Biomarker for Preeclampsia	2020	36	Maternal peripheral blood	Digital droplet PCR	Circulating C19MC microRNAs have a role in the etiology of pre-eclampsia
Jin, Yan et al. [8]	The Predictive Value of microRNA in Early Hypertensive Disorder Complicating Pregnancy (HDCP)	2021	136	Maternal peripheral blood	Fluorescence quantitative PCR	Pre-eclampsia with a clinically confirmed diagnosis is linked to changes in extracellular microRNA expression
Jing, Jia et al. [9]	Maternal Obesity alters C19MC microRNAs Expression Profile In Fetal Umbilical Cord Blood	2020	66	Fetal umbilical cord blood samples	Real-time PCR	According to miRNA profiling using the high-throughput Open Array TM technology, seven microRNAs have a distinct abundance profile in early pre-eclampsia
Légaré, Cécilia et al. [10]	First Trimester Plasma microRNAs Levels Predict Matsuda Index-Estimated Insulin Sensitivity Between 24th And 29th Week of Pregnancy	2022	421	Plasma samples	PCR	Up-regulation of circulating C19MC microRNAs is a hallmark of early pregnancy, which is predisposed to developing issues linked to gestational hypertension and placental insufficiency Elevated plasmatic levels of miR-516-5p, miR-518b, and miR-520h in the first trimester alone are strong indicators of future gestational hypertension
Li, Hui et al. [11]	Unique MicroRNA Signals In Plasma Exosomes from Pregnancies Complicated By Preeclampsia	2020	60	Maternal peripheral blood	Real-time PCR	Females who subsequently had severe pre-eclampsia had upregulated placental-specific miR-520a in their sera at 12–14 weeks of pregnancy
Lv, Yan et al. [12]	Roles of microRNAs in Preeclampsia	2019	No information	Various	Various	Circulating C19MC microRNAs have a role in pre-eclampsia
Miura, Kiyonori et al [13]	Circulating Chromosome 19 miRNA Cluster microRNAs In Pregnant Women with Severe Pre-Eclampsia	2015	40	Maternal peripheral blood	Real-time PCR	Changes in extracellular microRNA expression are linked to clinically confirmed hypertension and pre-eclampsia
Munjas, Jelena et al. [14]	Non-Coding RNAs in Preeclampsia—Molecular Mechanisms and Diagnostic Potential	2021	No information	Various	Various	Females who subsequently had severe pre-eclampsia had upregulated placental-specific microRNAs in their sera at 12–14 weeks of pregnancy
Murakami, Yuko et al. [15]	Reference Values For Circulating Pregnancy-Associated Micrornas in Maternal Plasma and their Clinical Usefulness in Uncomplicated Pregnancy and Hypertensive Disorder of Pregnancy	2018	33	Maternal peripheral blood	Real-time PCR	Four of the fifteen of the C19MC microRNAs that were examined showed down-regulation in placental tissues when gestational hypertension patients were present
Oostdam, Herrera-Van et al. [16]	Placental Exosomes Isolated from the Urine of Patients with Gestational Diabetes Exhibit a Differential Profile Expression of microRNAs across Gestation	2020	61	Maternal urine samples	Real-time PCR	Pregnancy-related difficulties are caused by several pathological and physiological processes in which microRNAs play a vital role
Qin, Shiting et al. [17]	The Value of Circulating microRNAs for Diagnosis and Prediction of Pre-eclampsia: A Meta-analysis and Systematic Review	2021	4727	Various	Various	The up-regulation of the C19MC microRNAs is a defining feature of pre-eclampsia that has already developed
Špačková, Kamila [2]	First-trimester Screening of Pregnancy-Related Complications Using Plasma Exosomal C19MC microRNAs	2019	97	Maternal peripheral blood	Real-time PCR	Several hypoxia-regulated microRNAs were complicated by extremely preterm fetal growth restriction Used microarray analysis to find 19 mature miRNAs that were differentially expressed in the blood of pregnant women who later had acute pre-eclampsia
Srinivasan, Srimeenakshi et al. [18]	Discovery and Verification of Extracellular miRNA Biomarkers for Non-Invasive Prediction of Pre-Eclampsia in Asymptomatic Women	2020	1097	Various	Various	Extracellular C19MC microRNAs are capable of distinguishing between individuals at risk of later developing placental sufficiency-related issues and normal pregnancies at the onset of gestation
Ura, Blendi et al. [19]	Potential Role of Circulating Micrornas as Early Markers of Pre-eclampsia	2014	48	Maternal peripheral blood	Real-time PCR	Circulating C19MC microRNAs may contribute to developing pre-eclampsia and prenatal hypertension in early pregnancy
Whigham, Carole-Anne et al. [20]	MicroRNAs 363 and 149 are Differentially Expressed in the Maternal Circulation Preceding a Diagnosis of Pre-eclampsia	2020	46	Maternal peripheral blood	Real-time PCR	Elevated plasma levels of microRNAs were seen in the group of participants with developed pre-eclampsia
Wommack, Joel C., et al. [21]	Micro RNA Clusters in Maternal Plasma are Associated with Preterm Birth and Infant Outcomes	2018	42	Maternal peripheral blood	Real-time PCR	miRNAs act as signaling molecules that coordinate infant outcomes and length of gestation

**Table 2 jcm-11-07051-t002:** Cochrane Bias Risk Assessment.

Study	(1)	(2)	(3)	(4)	(5)	(6)	(7)
Ali, Asghar et al. [3]	+	+	-	-	?	+	+
Cronqvist, Tina et al. [4]	+	+	-	+	+	+	?
Demirer, Selin et al. [5]	+	+	-	+	+	+	+
He, Xin, and Dan-Ni Ding [6]	+	+	-	-	?	+	+
Hromadnikova, Ilona et al. [1]	+	+	-	-	+	+	?
Jelena, Munjas et al. [7]	+	+	-	+	+	+	+
Jin, Yan et al. [8]	+	+	-	+	+	+	?
Jing, Jia et al. [9]	+	+	-	-	?	+	+
Légaré, Cécilia et al. [10]	+	+	+	+	+	+	+
Li, Hui et al. [11]	+	+	+	+	+	+	+
Lv, Yan et al. [12]	+	+	+	+	+	+	+
Miura, Kiyonori et al. [13]	+	+	+	+	?	+	+
Munjas, Jelena et al. [14]	+	+	-	-	+	+	?
Murakami, Yuko et al. [15]	+	+	+	-	+	+	+
Oostdam, Herrera-Van et al. [16]	+	+	-	+	+	+	+
Qin, Shiting et al. [17]	+	+	-	-	+	+	+
Špačková, Kamila [2]	+	+	-	+	?	+	?
Srinivasan, Srimeenakshi et al. [18]	+	+	+	+	+	+	?
Ura, Blendi et al. [19]	+	+	+	-	+	+	+
Whigham, Carole-Anne et al. [20]	+	+	-	+	+	+	+
Wommack, Joel C., et al. [21]	+	+	-	+	+	+	+

(1) Selection Bias (Random Sequence Generation); (2) Selection Bias (Allocation Concealment); (3) Performance Bias; (4) Detention Bias; (5) Attrition Bias; (6) Reporting Bias; and (7) Other Bias. Symbols: unclear risks (?), high risk (-), and low risk (+).

## Data Availability

Not applicable.

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
