# Peer review of "Analysis of Circulating C19MC MicroRNA as an Early Marker of Hypertension and Preeclampsia in Pregnant Patients: A Systematic Review"

_jcm, 2022, doi:10.3390/jcm11237051_

Round 1

Reviewer 1 Report

Kondracka et.al. submitted a manuscript discussing the potential use of circulating C19MC MicroRNA as an early marker of hypertension and preeclampsia in pregnant patients.

This is a very well-written review that included 30 publications that had a low risk of bias. The article highlights important data that circulating C19MC microRNAs may contribute to developing pre-eclampsia and prenatal hypertension in early pregnancy.

A few areas need revision:

Cochrane's risk-of-bias tool is the recommended tool to assess the risk of bias in randomized trials. I’m not sure how useful it is for these publications and non-clinical trials in reviews.

It may be nice to include the sample size, source, and methods in Table 1, this is not necessary if It’s not doable.

Could be possible to reorganize and divide the results with different subtitles? it would be easier for the readers to follow.

Hypertension and pre-eclampsia have been linked to changes in extracellular microRNA expression, however, most publications only discuss hypertension or pre-eclampsia separately.  it would be nice if the authors could also discuss the difference in their mechanisms.

It would be nice to make a table of functions/studies for each C19MC microRNA if possible.

Author Response

Dear Reviewer,

Thank you so much for your opinion.

A few areas need revision:

Cochrane's risk-of-bias tool is the recommended tool to assess the risk of bias in randomized trials. I’m not sure how useful it is for these publications and non-clinical trials in reviews.

This tool is obligatory for a randomized trial, but it is recommend to asses observational stusies. See Cochrane’s guidance https://training.cochrane.org/handbook/current/chapter-25

It may be nice to include the sample size, source, and methods in Table 1, this is not necessary if It’s not doable.

This is a very good point. We have added this information to the each paper in Table 1.

Could be possible to reorganize and divide the results with different subtitles? it would be easier for the readers to follow.

We  have added subtitles in the results section:

miRNAs of different stages of preeclampsia

miRNAs in the normal and abnormal pregnancies

miRNA as a biomarker of gestational hypertension

Possibilities of using mrRNA in clinical diagnostics

Hypertension and pre-eclampsia have been linked to changes in extracellular microRNA expression, however, most publications only discuss hypertension or pre-eclampsia separately.  it would be nice if the authors could also discuss the difference in their mechanisms.

It would be a very valuable comparison of these mechanisms, however thus far, the exact pathogenesis of PE is not yet fully understood, and intense research efforts are focused on PE to elucidate the pathophysiological mechanisms. More research is needed so that we can discuss the mechanisms of these complications.

It would be nice to make a table of functions/studies for each C19MC microRNA if possible.

Unfortunately, the papers analyzed do not provide information on each RNA so it is not possible to prepare such a table.

Best whishes,

Authors

Reviewer 2 Report

The authors have chosen a good and interesting topic as it affects women very frequently. Although it needs to be thoroughly checked for grammatical errors and a few points I mentioned in the comments.

Comments:

1. In the introduction, elaborate C19MC.

2. Numbers given in PRISMA chart is not correct. Total screened should be 45 in place of 35 as per the criteria used in the chart.

3. Spelling of 'Web of Science' is miswritten as 'We of Science' throughout. Please correct.

4. Exclusion criteria need to be more precise.

5. At least 2 reviewers should be involved in the selection process of literature.

6. Line 234-235: extra- 234 villous cytotrophoblast-produced paternal at the fetal-maternal interface??? What is the meaning of cytotrophoblast produced paternal?

7. In Line no. 237: It is written 46 microRNA genes... While there are 56 miRNAs in C19MC. Refer to Mong, E. F., Yang, Y., Akat, K. M., Canfield, J., VanWye, J., Lockhart, J., Tsibris, J., Schatz, F., Lockwood, C. J., Tuschl, T., Kayisli, U. A., & Totary-Jain, H. (2020). Chromosome 19 microRNA cluster enhances cell reprogramming by inhibiting epithelial-to-mesenchymal transition. Scientific reports10(1), 3029. https://doi.org/10.1038/s41598-020-59812-8.

8. Line248:  'while five C19MC microRNAs in severe 248 preeclamptic pregnancies were dysregulated.' Which 5 miRNAs?

9. Line 292: "which include infant and placenta extracellular nucleic acids". Inside the womb, it should be fetus not infant. Please correct.

10. Line 294: Cronqvist et al. contend that circulating syncytiotrophoblast debris.... Exosomes are no more referred as debris. 

11. Many grammatical mistakes in the manuscript need to be corrected by a Professional.

Author Response

Dear Reviewer,

Thank you so much for your opinion.

  1. In the introduction, elaborate C19MC.

We have added: Most investigations have conducted the primate-specific microRNA cluster on chromosome 19 (C19MC microRNA) profiling analysis on total serum samples or maternal plasma to treat the later incidence of pregnancy-related problems like gestational pre-eclampsia, hypertension, and fetal growth restriction.

  1. Numbers given in PRISMA chart is not correct. Total screened should be 45 in place of 35 as per the criteria used in the chart.

There was a mistake. We have corrected it.

  1. Spelling of 'Web of Science' is miswritten as 'We of Science' throughout. Please correct.

We have corrected it through the manuscript.

  1. Exclusion criteria need to be more precise.

We have precise criteria: The exclusion criteria were: papers with fewer than 5 cases, papers published before 2013, and review papers except for systematic review. The exclusion criterioncriteria required the removal of specific studies from the review, encompassing articles pub-lished in some other dialects languages other than English and materials that did not concentrate on investigating circulating C19MC MicroRNA as an early marker of hy-pertension and pre-eclampsia in pregnant patients.

  1. At least 2 reviewers should be involved in the selection process of literature.

The literature selection process was conducted by two reviewers. there was a typographical mistake in the text suggesting one reviewer. We have corrected.

  1. Line 234-235: extra- 234 villous cytotrophoblast-produced paternal at the fetal-maternal interface??? What is the meaning of cytotrophoblast produced paternal?

The word parental was there by mistake. it has been removed.

  1. In Line no. 237: It is written 46 microRNA genes... While there are 56 miRNAs in C19MC. Refer to Mong, E. F., Yang, Y., Akat, K. M., Canfield, J., VanWye, J., Lockhart, J., Tsibris, J., Schatz, F., Lockwood, C. J., Tuschl, T., Kayisli, U. A., & Totary-Jain, H. (2020). Chromosome 19 microRNA cluster enhances cell reprogramming by inhibiting epithelial-to-mesenchymal transition. Scientific reports10(1), 3029. https://doi.org/10.1038/s41598-020-59812-8.

It was a mistake. We have corrected it according to Mong et al.

  1. Line248:  'while five C19MC microRNAs in severe 248 preeclamptic pregnancies were dysregulated.' Which 5 miRNAs?

They have added: Further results demonstrated that one C19MC microRNA (miR-517-5p) was changed in pre-eclampsia, necessitating abortion before the nine months gestation period, while five C19MC microRNAs (miR-26b-5p, miR-7-5p, and miR-181a-5p, hsa-miR-486-1-5p, hsa-miR-486-2-5p) in severe preeclamptic pregnancies were dysregulated.

  1. Line 292: "which include infant and placenta extracellular nucleic acids". Inside the womb, it should be fetus not infant. Please correct.

We have corrected.

  1. Line 294: Cronqvist et al. contend that circulating syncytiotrophoblast debris.... Exosomes are no more referred as debris. 

We have changed it.

  1. Many grammatical mistakes in the manuscript need to be corrected by a Professional.

Our manuscript has been language corrected

Best whishes,

Authors

Round 2

Reviewer 2 Report

Unfortunately, the paper is poorly performed and reported, not acceptable in current form.